# Cholesin mRNA Expression in Human Intestinal, Liver, and Adipose Tissues

**DOI:** 10.3390/nu17040619

**Published:** 2025-02-08

**Authors:** Hannah Gilliam-Vigh, Malte P. Suppli, Sebastian M. N. Heimbürger, Asger B. Lund, Filip K. Knop, Anne-Marie Ellegaard

**Affiliations:** 1Center for Clinical Metabolic Research, Copenhagen University Hospital—Herlev and Gentofte, DK-2900 Hellerup, Denmark; 2Department of Biomedical Sciences, Faculty of Health and Medical Sciences, University of Copenhagen, DK-2200 Copenhagen, Denmark; 3Clinical Research, Steno Diabetes Center Copenhagen, DK-2730 Herlev, Denmark; 4Department of Clinical Medicine, The Faculty of Health and Medical Sciences, University of Copenhagen, DK-2200 Copenhagen, Denmark

**Keywords:** cholesin, cholesterol, mRNA, white adipose tissue, intestines, liver

## Abstract

Objective: Cholesin is a recently discovered gut-derived hormone secreted by enterocytes upon dietary cholesterol uptake via the transmembrane sterol transporter Niemann–Pick disease C1-like intracellular cholesterol transporter 1 (NPC1L1). In the liver, cholesin activates G protein-coupled receptor 146 (GPR146), causing reduced cholesterol synthesis. In this exploratory, hypothesis-generating study based on post hoc analysis, human data on the cholesin system are presented. Methods: Mucosal biopsies were collected throughout the intestinal tract from 12 individuals with type 2 diabetes (T2D) and 12 healthy, matched controls. Upper small intestinal mucosal biopsies were collected from 20 individuals before and after Roux-en-Y gastric bypass (RYGB) surgery. Liver biopsies were collected from 12 men with obesity and 15 matched controls without obesity. Subcutaneous abdominal adipose tissue biopsies were collected from 20 men with type 1 diabetes (T1D). All biopsies underwent full mRNA sequencing. Results: Cholesin mRNA expression was observed throughout the intestinal tracts of the individuals with T2D and the controls, in the livers of men with and without obesity, and in adipose tissue of men with T1D. NPC1L1 mRNA expression was robust throughout the small intestines but negligible in the large intestines of both individuals with and without T2D. RYGB surgery induced the expression of NPC1L1 mRNA in the upper small intestine. GPR146 mRNA was expressed in the livers of men, both with and without obesity, and in the adipose tissue of men with T1D, but not in the intestines. Conclusions: Our results suggest a role of the cholesin system in human physiology, but whether it is perturbed in metabolic diseases remains unknown. Clinical trial registration numbers: NCT03044860, NCT03093298, NCT02337660, NCT03734718.

## 1. Introduction

Cholesterol homeostasis is maintained through an intricate balance between the absorption of dietary cholesterol, hepatic cholesterol synthesis, fecal excretion, and utilization of cholesterol as a substrate in the synthesis of other compounds, such as bile acids and steroid hormones [1]. Regulating this balance is important to control the level of circulating cholesterol. Elevated levels of circulating cholesterol, especially in the form of low-density lipoprotein (LDL) cholesterol, is a contributing factor to the development of atherosclerosis and is associated with an increased risk of cardiovascular disease [2].

Recently, a novel gut-derived hormone, cholesin (encoded by chromosome 7, open reading frame 50 [c7orf50]), was discovered and shown to regulate cholesterol homeostasis through the gut–liver axis [3]. It was further demonstrated that cholesin is expressed and secreted by enterocytes upon cholesterol uptake from the gut lumen via Niemann–Pick disease C1-like intracellular cholesterol transporter 1 (NPC1L1) [3]. Cholesin then travels via circulation to its target organs, which in mice, include the liver, white and brown adipose tissue, kidney, and skeletal muscle [3]. Through activation of the cholesin receptor (G protein-coupled receptor 146 (GPR146)), a cellular signaling cascade involving protein kinase A, cyclic AMP, and partly extracellular signal-regulated kinase 1/2 facilitates the reduced expression of cholesterogenic genes, and hence, the reduction of circulating cholesterol levels [3]. In mice, long-term consequences of this reduction include a dramatic reduction of atherosclerotic lesions, especially in combination with the cholesterol synthesis-inhibiting drug rosuvastatin, in an atherosclerosis-prone mouse model [3]. As such, this novel gut–liver axis, which controls cholesterol homeostasis, constitutes an important new therapeutic target for improving the management of hypercholesterolemia and atherosclerosis [3].

The homology between mouse and human cholesin is only 64%, suggesting some degree of inter-species heterogeneity. Hu et al.’s in vitro and mouse experiments were supported by negative correlations of plasma cholesin levels with plasma levels of total cholesterol, LDL cholesterol, triglycerides, and apolipoprotein B in humans [3]. Furthermore, two single-nucleotide polymorphisms (rs10236293 and rs1007765, both located near the 3′ end of the c7orf50 gene) were found to be associated with the levels of plasma cholesin, total cholesterol, and LDL cholesterol in human individuals [3]. Also, recent research by Ryk et al. showed that GPR146 mRNA expression in epicardial adipose tissue was associated with body mass index (BMI) and plasma concentrations of total cholesterol and LDL in individuals with type 2 diabetes, whereas no association was observed in healthy controls [4]. Additionally, they found a set of genes coexpressed with GPR146, including key regulators of lipid and fatty acid metabolism, which was downregulated during selective sodium glucose cotransporter 2 inhibitor treatment [4]. Collectively, these findings indicate that the cholesin system plays a role in human metabolic physiology.

Here, with a hypothesis-generating aim, we report the mRNA expressions of c7orf50, NPC1L1, and GPR146 in biobanked samples from previous studies, including mucosa biopsies collected along the entire intestinal tract, as well as liver and adipose tissue biopsies from healthy individuals and/or in individuals with different metabolic conditions.

## 2. Methods

### 2.1. Ethical Approval

The studies were approved by the Scientific Ethical Committee of the Capital Region of Denmark (registration numbers H-3-2010-115 (appr. date: 1 December 2010), H-6-2014-047 (appr. date: 3 July 2015), H-6-2014-097 (appr. date: 18 December 2014), and H-18015379 (appr. date: 23 August 2018)) and the Danish Data Protection Agency. All studies were registered with clinicaltrials.gov (NCT03044860, NCT03093298, NCT02337660, and NCT03734718). The studies comply with the most recent revision of the Declaration of Helsinki, and informed consents, both oral and written, were obtained from all participants before their enrollment. Details of the clinical experimental procedures have been described in previous publications [5,6,7,8].

### 2.2. Study Participants

Rhee et al. [5] collected gut mucosal biopsies from patients with type 2 diabetes and healthy controls matched for sex, age, and BMI. To be included, individuals with type 2 diabetes needed to have been diagnosed at least three months prior and be managing their condition through lifestyle counseling, metformin, or sulfonylurea. The exclusion criteria specified the use of other glucose-lowering drugs, a BMI greater than 35 kg/m^2^, and contraindications to sedation or enteroscopy [5]. Small intestinal mucosal biopsies were collected by Jorsal et al. [6] before and after Roux-en-Y gastric bypass (RYGB) surgery. The inclusion criteria were a referral for RYGB, which is based on BMI ≥ 50 or BMI ≥ 35 kg/m^2^ and at least one obesity-related comorbidity, and the exclusion criteria included chronic infections, inflammatory diseases, and recent corticosteroid use [6]. Liver biopsies were collected by Suppli et al. [7] from individuals with obesity and healthy lean individuals. The exclusion criteria included a history of liver disease, weekly alcohol consumption exceeding 14 units, diabetes, and first-degree relatives with diabetes [7]. Abdominal adipose tissue biopsies were collected by Heimbürger et al. [8] from lean men with type 1 diabetes. The inclusion criteria included a type 1 diabetes duration of 2–15 years, and glycated hemoglobin (HbA1c) < 69 mmol/mol (8.5%). The exclusion criteria included treatment with glucose-lowering drugs other than insulin, reduced kidney function, and known liver disease [8]. Detailed inclusion and exclusion criteria are described in the original articles [5,6,7,8]. The participant characteristics can be found in Table 1.

### 2.3. Biopsy Sampling

Details on biopsy sampling were previously described [5,6,7,8]. In short, intestinal samples from the individuals with type 2 diabetes and their matched controls were collected during anterograde and retrograde double-balloon enteroscopies performed using an EN-450 T5 enteroscope (Fujinon Inc., Tokyo, Japan) [5]. During the anterograde enteroscopy, the maximal insertion depth was marked with ink, and mucosal biopsies were taken at 30 cm intervals from the jejunum, ileum, ligament of Treitz, and duodenum. In the subsequent retrograde session, biopsies were collected from the marked depth at 30 cm intervals throughout the small intestine and from the ileocecal region, cecum, the ascending, the transverse, the descending, and the sigmoid colon, as well as from the rectum. The biopsy samples from each site were immediately immersed in RNAlater solution (Sigma Aldrich R0901, St. Louis, MO, USA). Before RYGB, mucosal samples were collected from the small intestine at the expected site of jejuno–jejunal anastomosis [6]. Approximately three months after RYGB, biopsy specimens were obtained from the alimentary limb, the biliopancreatic limb, and the common channel. The biopsy specimens were immediately embedded in Tissue-Tek OCT Compound (Sakura Finetek, Torrance, CA, USA). Liver tissue samples from participants, both with and without obesity, were collected by an ultrasound-guided liver biopsy [7]. This procedure was performed by a radiologist using a BARD MONOPTY Disposable Core Biopsy Instrument (Becton Dickinson, Franklin Lakes, NJ, USA), and the biopsies were placed in RNAlater (Thermo-Fisher Scientific, Waltham, MA, USA). For individuals with and without type 1 diabetes, adipose tissue biopsies were collected and snap-frozen in liquid nitrogen [8].

### 2.4. Tissue Handling and mRNA Analyses

Detailed methodologies are described in the original articles [5,6,7,8]. mRNA sequencing was performed on various tissue samples, including biopsies from the intestinal mucosa, liver, and abdominal adipose tissue. The general procedure involved tissue homogenization, RNA isolation, and quality assessment. Tissue homogenization was achieved with a TissueLyzer (Qiagen, Hilden, Germany) using 5 mm steel beads. Total RNA was isolated from the homogenate using the NucleoSpin RNA kit (Macherey-Nagel, Düren, Germany). The quantity of the purified RNA was measured using the Qubit RNA BR Assay Kit (Thermo Fisher Scientific, Waltham, MA, USA). The quality of the purified RNA was assessed with a bioanalyzer using the Agilent RNA 6000 Nano Kit (Agilent Technologies, Waldbronn, Germany). Subsequently, complementary DNA (cDNA) libraries were prepared with 25 to 100 ng of purified RNA samples using the TruSeq Stranded mRNA Library Prep Kit for NeoPrep (Illumina, San Diego, CA, USA). Sequencing of the cDNA libraries was performed with the NextSeq 500 High Output Kit v2 (75 cycles) (Illumina) on the NextSeq 500 platform. All mRNA sequencing analyses were performed by Gubra A/S (Hørsholm, Denmark). The gene expressions are reported in the figures and tables as reads per kilobase of transcript per million mapped reads (RPKM). Expression levels above 1 RPKM are considered to reflect robust expression, while the threshold for negligible expression was set at 0.1 RPKM.

### 2.5. Data Analysis and Statistical Analyses

The clinical datasets were summarized using counts or mean values, supplemented by corresponding ranges. Box and whisker plots and scatter plots were used for data visualization. Human tissue samples were analyzed using full mRNA sequencing, revealing differences in the mRNA expression levels of NPC1L1, c7orf50, and GPR146. A linear mixed-effects model was employed to compare individuals with type 2 diabetes and healthy controls across various regions of the small and large intestines, with group and location as the fixed effects, as well as the interaction between these factors. To account for repeated measurements within individuals, an unstructured covariance pattern was applied. Site-specific differences between individuals with type 2 diabetes and healthy controls were identified, along with inter-site variations, using the duodenum and rectum as reference points for the small and large intestines, respectively. Moreover, gut mucosal samples were analyzed to detect differences in the mRNA expression levels between biopsies taken before and after RYGB surgery across three intestinal locations. Similarly, a linear mixed-effects model was applied, focusing on location as the fixed effect, with an unstructured covariance pattern to handle repeated measures. For comparisons between individuals with obesity and lean individuals, unpaired t tests were used. Only anatomical sites where more than 50% of the participants exhibited non-negligible expression levels (>0.1 RPKM) were included in the analyses. The small intestine, large intestine, RYGB operated intestine, and liver were considered independently. Due to non-normal distributions, all data were log-transformed, and the results are presented as the relative median differences with 95% confidence intervals. To account for multiple comparisons, the Benjamini–Hochberg method was used to control the false discovery rate. A corrected *p*-value below 0.05 was considered statistically significant. Analyses were performed using SAS software (SAS Studio version 3.8), and figures were produced using R Studio [9] (Version 2022.07.2 + 576). The statistical approach aligns with the original study protocols [5,6,7,8] but introduces a refined statistical methodology.

## 3. Results

### 3.1. Nieman–Pick Disease C1-like Intracellular Cholesterol Transporter 1 (NPC1L1)

In both healthy individuals and those with type 2 diabetes, NPC1L1 was highly expressed in the small intestine, with mean levels rising from the duodenum (69 ± 6.8 RPKM (mean ± SEM)) to location 6 (120 ± 7.3 RPKM), followed by a steep drop in the distal part of the small intestine (ileocecal: 1 ± 0.73 RPKM), and absence in the large intestine (Figure 1A, Appendix A). The expression pattern in individuals with type 2 diabetes was similar to that in healthy individuals except for a significant difference observed in location 8 (*p* = 0.01) (Figure 1A, Appendix A). However, as discussed previously [5], exact biopsy locations in the lower small intestine are associated with some degree of uncertainty, and therefore, comparisons between individuals with type 2 diabetes and healthy individuals in this region should be interpreted cautiously. After RYGB surgery, the mean expression of NPC1L1 at the alimentary limb was increased compared to its expression pre-operatively at the expected site of jejuno–jejunal anastomosis (200 ± 16 vs. 130 ± 7.6 RPKM, *p* = 0.003) (Figure 1B, Appendix A). The expression levels at the biliopancreatic limb and the common channel were similar to the pre-operative levels (Figure 1B, Appendix A). In the liver biopsies from men with and without obesity, the mean RPKM values for NPC1L1 amounted to 11 ± 0.68 and 13 ± 1.1 (*p* = 0.45) (Figure 1C, Appendix A). In the adipose tissue of individuals with type 1 diabetes, the expression of NPC1L1 was low (0.92 ± 0.19 RPKM) (Figure 1D).

### 3.2. Cholesin (c7orf50)

In individuals with and without type 2 diabetes, the expression of c7orf50 was robust throughout the intestines, with slightly higher levels in the large intestine (rectum: 5.2 ± 0.33 RPKM) compared to the small intestine (duodenum: 3.2 ± 0.2) (Figure 2A, Appendix A). Both groups exhibited similar expression patterns, with no significant differences between the groups (Appendix A). There was no significant effect of RYGB surgery on the expression of c7orf50 mRNA in the alimentary limb, biliopancreatic limb, or common channel compared to the pre-operation conditions (Figure 2B, Appendix A). C7orf50 expression in the liver biopsies from men with and without obesity was similar between the groups (3.2 ± 0.22 RPKM vs. 3.2 ± 0.16 RPKM, *p* = 0.96) (Figure 2C, Appendix A). The overall highest mean level of c7orf50 mRNA expression was found in the adipose tissue of individuals with type 1 diabetes (7.9 ± 0.32 RPKM) (Figure 2D).

### 3.3. G Protein-Coupled Receptor 146 (GPR146)

The mRNA expression level of GPR146 was very low (<1 RPKM in all locations) throughout the intestines of individuals with and without type 2 diabetes with similar expression patterns (Figure 3A, Appendix A); no significant differences between the duodenum and other small intestinal locations were observed after adjusting for multiple testing (Appendix A). Similarly, low levels of GPR146 expression were observed in the mucosal samples taken before and after RYGB (<1 RPKM in all locations). The expression levels did not show significant changes postoperatively compared to preoperatively (Appendix A). In the liver biopsies from men with and without obesity, the mean levels of GPR146 expression amounted to 5.7 ± 0.44 RPKM and 6 ± 0.32 (*p* = 0.82) (Figure 3C). No significant differences in GPR146 mRNA expression were observed between the men with and without obesity (Appendix A). The overall highest mean level of GPR146 mRNA expression was seen in the adipose tissue of individuals with type 1 diabetes (35 ± 1.2 RPKM) (Figure 3D).

## 4. Discussion

Based on the recent discovery of the gut-derived hormone cholesin, its receptor (GPR146), and its role in maintaining cholesterol homeostasis in mice, we investigated the mRNA expression of the cholesterol transporter NPC1L1, c7orf50, and GPR146 in human biopsies retrieved throughout the entire intestines of individuals with type 2 diabetes and matched healthy controls, from the livers of men with and without obesity, from the upper intestinal tract before and after the RYGB procedure, and from subcutaneous adipose tissue of men with type 1 diabetes. The cholesin gene c7orf50 was robustly expressed throughout the intestinal tracts of both individuals with type 2 diabetes and healthy controls, as well as in the livers of men with and without obesity. The highest expression was observed in the adipose tissue from men with type 1 diabetes. We report robust expression of NPC1L1 in the small intestines and negligible expression in the large intestines of individuals with and without type 2 diabetes. Furthermore, we observed increased NPC1L1 mRNA levels in the upper small intestine following RYGB surgery. The expression of GPR146 was high in the white adipose tissue and the liver.

One of the key findings by Hu et al. was the identification of cholesin expression in enterocytes, which requires cholesterol stimulation of the NPC1L1 in these cells [3]. In microscopic images of intestinal sections from genetically manipulated mice, the overlap between the enterocyte marker APOA and cholesin is evident, leading to the conclusion that cholesin is expressed in NPC1L1-expressing enterocytes in mice [3]. Interestingly, in our human dataset, which describes NPC1L1 mRNA expression and c7orf50 mRNA expression profiles along the entire human intestine from the duodenum to the rectum, there seems to be a spatial separation between c7orf50 mRNA expression and NPC1L1 mRNA expression, with negligible NPC1L1 mRNA levels in the large intestine, whereas c7orf50 mRNA expression was greatest in the large intestine.

Individuals with type 2 diabetes often suffer from dyslipidemia, with low levels of high-density lipoprotein cholesterol and high levels of triglycerides in the blood [10]. In the present dataset, the mRNA expression levels of NPC1L1, c7orf50, and GPR146, respectively, were similar in the individuals with type 2 diabetes compared with the age-, sex-, and BMI-matched healthy individuals. The differences observed among locations 7–9 most likely arise due to sampling differences in the two groups due to differences in small intestinal length, as discussed elsewhere [5]. Therefore, the present findings do not point to alterations in the cholesin system in individuals with type 2 diabetes.

Interestingly, we observed an increase in the small intestinal mRNA expression of NPC1L1 after RYGB compared to before surgery. A previous study found decreased cholesterol absorption after RYGB [11]. Whether NPC1L1 expression is influenced by cholesterol exposure and/or absorption remains, to our knowledge, unknown. However, the generally high expression of NPC1L1 mRNA in the small intestine and relatively low expression in the liver and adipose tissue is in line with the gut-derived cholesterol regulation by the cholesin system proposed by Hu et al. [3]. The low expression level of c7orf50 in all tissues might reflect the fasting state of the participants, which was needed for the biopsy procedures. To evaluate cholesin in a fed state, other methods or study designs are warranted.

In accordance with the findings by Hu et al. in mice, our mRNA data in humans suggest that GPR146 is not expressed in the intestines but robustly expressed in both the liver and white adipose tissue. The hepatic expression is in line with Hu et al.’s findings showing cholesin-induced inhibition of hepatic de novo cholesterol synthesis [3]. The role of cholesin in abdominal adipose tissue remains unknown; however, as described by Ryk et al. in epicardial adipose tissue, the cholesin system seems to be involved with systemic cholesterol metabolism in individuals with type 2 diabetes but not in healthy controls [4]. Clinical studies addressing the effects of cholesin and GPR146 activation in various tissues are warranted to understand the systemic effects of cholesin in human physiology and pathophysiology.

The uniform sampling, storage, and processing of the biopsies across the studies is a strength of our study. Furthermore, the tissues we investigated here include the most relevant according to the tissue expression of cholesin in mice [3]. However, several limitations must be acknowledged. First, our study is limited by the fact that only mRNA data were available and, therefore, protein levels cannot be deduced here. Second, the present data were derived from post hoc analyses and should be regarded as exploratory. The biopsy data were derived from different participant populations, including healthy individuals and individuals with type 1 diabetes, type 2 diabetes, and/or obesity, which complicates direct comparisons. Particularly, the lack of adipose tissue data from a healthy control group is important. Although the robust expressions of c7orf50 and GPR146 in the adipose tissue of individuals with type 1 diabetes suggest a potential role of the cholesin system in adipose tissue, these findings must be interpreted with caution without data from an appropriate control group. At this point, it is not possible to determine whether these observations reflect type 1 diabetes pathophysiology or general adipose tissue physiology. Dedicated studies to investigate this are warranted. Furthermore, while this study provides valuable insights into mRNA expression patterns, it is important to note that the potential confounding effects of diabetes treatments, such as metformin, sulfonylureas, and insulin, on mRNA expression cannot be entirely ruled out. Future studies designed to specifically assess the impact of these treatments would help clarify their potential influence. Lastly, all participants, from whom biopsies were obtained, were of Caucasian descent, and therefore, whether other ethnicities would show similar results is uncertain.

## 5. Conclusions

In conclusion, we report robust mRNA expression of key elements of the cholesin system—NPC1L1, c7orf50, and GPR146—in intestinal, liver, and adipose tissues in humans. These data indicate a role of the cholesin system in human physiology. Whether the cholesin system is involved in the pathophysiology of metabolic conditions remains uncertain.

## Figures and Tables

**Figure 1 nutrients-17-00619-f001:**
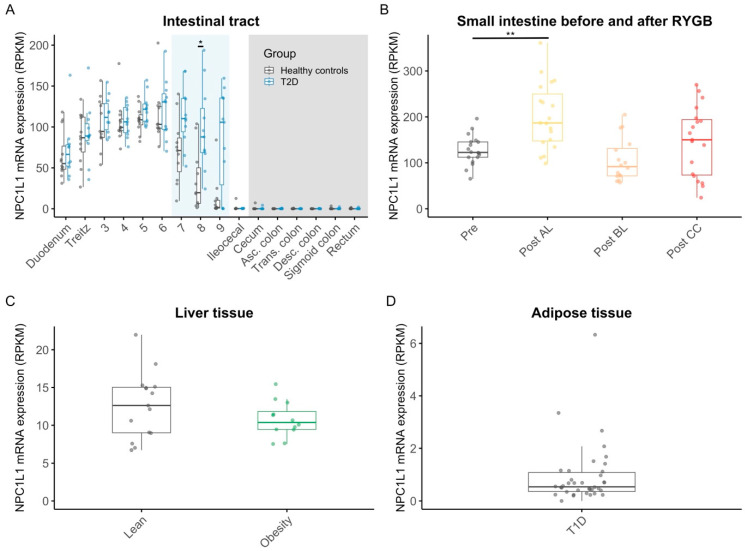
mRNA expression levels of NPC1L1 in the intestine, in the intestine before and after RYGB, in the liver, and in adipose tissue. mRNA expressions of NPC1L1 in mucosal biopsies sampled throughout the small intestine (white background/light blue background (light blue indicates sections of the intestine where the exact locations of the biopsies were taken with considerable uncertainty)) and the large intestine (grey background) in 12 individuals with type 2 diabetes (blue) and in 12 age- and body-mass-index-matched healthy controls (grey) (**A**); in small intestinal mucosal samples from 19 individuals collected after RYGB in the alimentary limb (yellow), biliopancreatic limb (orange), common channel (red), and before RYGB (grey) (**B**); in transcutaneously sampled liver biopsies from 12 men with obesity (green) and 15 lean controls (grey) (**C**); in subcutaneous adipose tissue biopsies from 20 men with type 1 diabetes (grey) (**D**). Dots are individual data points; boxes represent inter-quartile ranges, and whiskers extend from the 25th percentile to the smallest value within 1.5 times the interquartile range below it and from the 75th percentile to the largest value within 1.5 times the interquartile range above it (encompassing data points not deemed outliers). Statistical significance is represented as follows: ** for *p* < 0.01, * for *p* < 0.05. For non-significant results (*p* ≥ 0.05), the *p* values are not displayed. Abbreviations: Asc., ascending; Desc., descending; NPC1L1, Niemann–Pick disease C1-like intracellular cholesterol transporter 1; Post AL, postsurgery alimentary limb; Post BL, postsurgery biliopancreatic limb; Post CC, postsurgery common channel; Pre, presurgery; RPKM, reads per kilobase of transcript per million mapped reads; Trans., transverse.

**Figure 2 nutrients-17-00619-f002:**
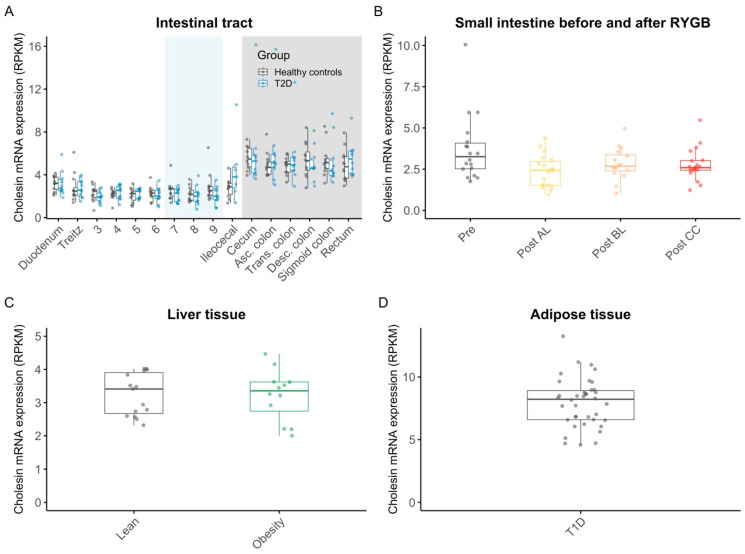
mRNA expression levels of cholesin in the intestine, in the intestine before and after RYGB, in the liver, and in the adipose tissue. mRNA expression of cholesin in mucosal biopsies sampled throughout the small intestine (white background/light blue background (light blue indicates sections of the intestine where the exact locations of the biopsies were taken with considerable uncertainty)) and the large intestine (grey background) in 12 individuals with type 2 diabetes (blue) and in 12 age- and body-mass-index-matched healthy controls (grey) (**A**); in small intestinal mucosal samples from 19 individuals collected after RYGB in the alimentary limb (yellow), biliopancreatic limb (orange), common channel (red), and before RYGB (grey) (**B**); in transcutaneously sampled liver biopsies from 12 men with obesity (green) and 15 lean controls (grey) (**C**); in subcutaneous adipose tissue biopsies from 20 men with type 1 diabetes (grey) (**D**). Dots are individual data points; boxes represent inter-quartile ranges, and whiskers extend from the 25th percentile to the smallest value within 1.5 times the interquartile range below it and from the 75th percentile to the largest value within 1.5 times the interquartile range above it (encompassing data points not deemed outliers). Abbreviations: Asc., ascending; Desc., descending; Post AL, postsurgery alimentary limb; Post BL, postsurgery biliopancreatic limb; Post CC, postsurgery common channel; Pre, presurgery; RPKM, reads per kilobase of transcript per million mapped reads; Trans., transverse.

**Figure 3 nutrients-17-00619-f003:**
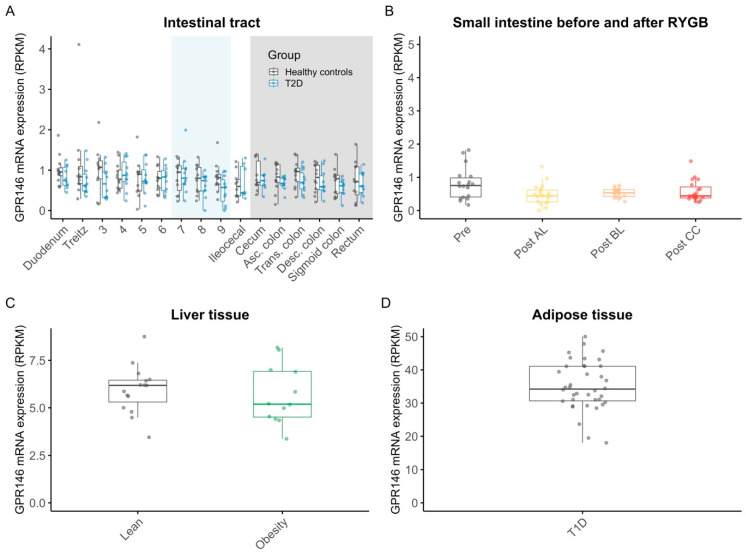
mRNA expression levels of GPR146 in the intestine, in the intestine before and after RYGB, in the liver, and in the adipose tissue. mRNA expression of GPR146 in mucosal biopsies sampled throughout the small intestine (white background/light blue background (light blue indicates sections of the intestine where the exact locations of the biopsies were taken with considerable uncertainty)) and the large intestine (grey background) in 12 individuals with type 2 diabetes (blue) and in 12 age- and body-mass-index-matched healthy controls (grey) (**A**); in small intestinal mucosal samples from 19 individuals collected after RYGB in the alimentary limb (yellow), biliopancreatic limb (orange), common channel (red), and before RYGB (grey) (**B**); in transcutaneous liver biopsies from 12 men with obesity (green) and 15 lean controls (grey) (**C**); in subcutaneously sampled adipose tissue biopsies from 20 men with type 1 diabetes (grey) (**D**). Dots are individual data points; boxes represent inter-quartile ranges, and whiskers extend from the 25th percentile to the smallest value within 1.5 times the interquartile range below it and from the 75th percentile to the largest value within 1.5 times the interquartile range above it (encompassing data points not deemed outliers). Abbreviations: Asc., ascending; Desc., descending; GPR146, G protein-coupled receptor 146; Post AL, postsurgery alimentary limb; Post BL, postsurgery biliopancreatic limb; Post CC, postsurgery common channel; Pre, presurgery; RPKM, reads per kilobase of transcript per million mapped reads; Trans., transverse.

**Table 1 nutrients-17-00619-t001:** Study participant demographics. Data are numbers or means with ranges in brackets. Groups were compared using unpaired *t* tests; * Individuals with type 2 diabetes were compared with healthy individuals; ^†^ individuals with overweight were compared to lean individuals. Abbreviations: BMI, body mass index; HbA1c, glycated hemoglobin; RYGB, Roux-en-Y gastric bypass.

	Study 1 [5]			Study 2 [6]	Study 3 [7]			Study 4 [8]
	Type 2 Diabetes	Healthy	*p* Value *	RYGB Patients	With Obesity	Without Obesity	*p* Value ^†^	Type 1 Diabetes
Participants	12	12		20	12	15		20
Sex ratio (M/F)	9/3	8/4	1.00	6/14	12/0	15/0	1.00	20/0
Age (years)	51 (34–63)	50 (41–67)	0.66	44 (29–56)	36 (25–58)	41 (25–68)	0.315	26 (18–49)
BMI (kg/m2)	27 (23–32)	27 (20–31)	0.92	43.3 (36–52)	33.6 (31–40)	23.2 (21–25)	<0.0001	23.8 (20–27)
HbA1c (mmol/mol)	48 (36–85)	34 (29–43)	0.008	37.9 (25–70)	31 (26–37)	30 (23–34)	0.34	51 (32–71)
Disease duration (years)	5 (1–9)	NA	NA	NA	NA	NA	NA	9.1 (2–15)

## Data Availability

The data will be made available from the corresponding author upon reasonable request. The data are not publicly available due to privacy concerns.

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
