# Peer review of "Cholesin mRNA Expression in Human Intestinal, Liver, and Adipose Tissues"

_nutrients, 2025, doi:10.3390/nu17040619_

Round 1

Reviewer 1 Report

Comments and Suggestions for Authors

The study titled “Cholesin mRNA expression in human intestinal, liver, and adipose tissues” by Gilliam-Vigh et al. represents a valuable contribution to the field, focusing on a relatively novel protein, cholesin, which has the potential to serve as a target for pharmaceutical interventions aimed at treating atherosclerotic plaques and hypercholesterolemia. Despite the limited sample size, the authors have included a diverse range of patient groups, such as individuals with type 1 diabetes (T1D), type 2 diabetes (T2D), obesity, and those who underwent Roux-en-Y gastric bypass surgery, and compared these to control groups regarding cholesin expression. This diversity enhances the generalizability of the findings.

Additionally, the authors examined the expression of the NPC1L1 cholesterol transporter and the receptor for cholesin, GPR146, providing a broader context for their findings. The limitations of the study are adequately addressed in the discussion section, reflecting a commendable level of transparency and allowing readers to appropriately weigh the study’s conclusions.

However, one important question arises: why were protein levels not assessed in the samples? Given that differences in mRNA expression do not always correlate with protein levels, this omission represents a missed opportunity to significantly enhance the study’s impact and provide a more comprehensive understanding of cholesin’s role.

Finally, it should be noted that while Novo Nordisk A/S is mentioned in the author affiliations, it should also be disclosed in an explicit manner in the conflict of interest section in order to adhere to ethical standards.

Author Response

Reviewer 1:

The study titled “Cholesin mRNA expression in human intestinal, liver, and adipose tissues” by Gilliam-Vigh et al. represents a valuable contribution to the field, focusing on a relatively novel protein, cholesin, which has the potential to serve as a target for pharmaceutical interventions aimed at treating atherosclerotic plaques and hypercholesterolemia. Despite the limited sample size, the authors have included a diverse range of patient groups, such as individuals with type 1 diabetes (T1D), type 2 diabetes (T2D), obesity, and those who underwent Roux-en-Y gastric bypass surgery, and compared these to control groups regarding cholesin expression. This diversity enhances the generalizability of the findings. Additionally, the authors examined the expression of the NPC1L1 cholesterol transporter and the receptor for cholesin, GPR146, providing a broader context for their findings. The limitations of the study are adequately addressed in the discussion section, reflecting a commendable level of transparency and allowing readers to appropriately weigh the study’s conclusions.

Comment 1: However, one important question arises: why were protein levels not assessed in the samples? Given that differences in mRNA expression do not always correlate with protein levels, this omission represents a missed opportunity to significantly enhance the study’s impact and provide a more comprehensive understanding of cholesin’s role.

AUTHOR RESPONSE: We agree that assessing protein levels would add important information to our study. However, due to the limited amount of tissue collected, it was not feasible to extract peptides for protein analysis.

We have attempted to address this in line 272 in the discussion “First, our study is limited by the fact that only mRNA data were available and, thus, protein levels cannot be deduced here.”

Comment 2: Finally, it should be noted that while Novo Nordisk A/S is mentioned in the author affiliations, it should also be disclosed in an explicit manner in the conflict of interest section in order to adhere to ethical standards

AUTHOR RESPONSE: We thank the reviewer for highlighting this oversight. The conflict of interest section has been updated to disclose the coauthor's affiliation with Novo Nordisk A/S explicitly.

Reviewer 2 Report

Comments and Suggestions for Authors

The study is a descriptive examination of mRNA expression of c7orf50, NPC1L1, and GPR146 in biobanked samples and in mucosa biopsies sampled along the entire intestinal tract as well as liver and adipose tissue biopsies from healthy individuals and/or in individuals with several different metabolic conditions. Overall, the study appears to be well conducted and statistically evaluated. The study provides a required baseline for future research on these target genes in disease states.

1.  Before publication the authors must include in a Table a list of forward and reverse 5’-3’ target primers and controls used for mRNA analysis of which includes the target gene primer sequence, length of amplicon (bp), and the GenBank accession number for purposes of reproducibility.

2. The authors should comment as a limitation on the potential confounding impact of drugs (metformin and sulfonylureas) on the mRNA targets examined in the T2D patients and insulin in the T1D patients. For example, it is entirely possible that treatment may have normalized the levels of mRNA expression of the targets examined.

Author Response

Reviewer 2.

The study is a descriptive examination of mRNA expression of c7orf50NPC1L1, and GPR146 in biobanked samples and in mucosa biopsies sampled along the entire intestinal tract as well as liver and adipose tissue biopsies from healthy individuals and/or in individuals with several different metabolic conditions. Overall, the study appears to be well conducted and statistically evaluated. The study provides a required baseline for future research on these target genes in disease states.

Comment 1: Before publication the authors must include in a Table a list of forward and reverse 5’-3’ target primers and controls used for mRNA analysis of which includes the target gene primer sequence, length of amplicon (bp), and the GenBank accession number for purposes of reproducibility.

AUTHOR RESPONSE: We thank the reviewer for their suggestion. However, this study utilized full mRNA sequencing rather than targeted qPCR; thus, no primer sequences or amplicon information are available. To ensure clarity, we have updated the manuscript in line 154: "Human tissue samples were analyzed using full mRNA sequencing, revealing differences in mRNA expression levels of NPC1L1, C7orf50, and GPR146."

Comment 2: The authors should comment as a limitation on the potential confounding impact of drugs (metformin and sulfonylureas) on the mRNA targets examined in the T2D patients and insulin in the T1D patients. For example, it is entirely possible that treatment may have normalized the levels of mRNA expression of the targets examined.

AUTHOR RESPONSE: We appreciate the reviewer’s observation. A statement has been added to the discussion to acknowledge the potential medicinal influence on mRNA expression levels at line 283: “Furthermore, while this study provides valuable insights into mRNA expression patterns, it is important to note that the potential confounding effects of diabetes treatments, such as metformin, sulfonylureas, and insulin, on mRNA expression cannot be entirely ruled out. Future studies designed to specifically assess the impact of these treatments would help clarify their potential influence.”

Round 2

Reviewer 2 Report

Comments and Suggestions for Authors

The revised version of the manuscript did not include the response change indicated for line 154 under the Tissue handling and mRNA analysis section. It appears erroneously in line 169. Please correct.

Author Response

Comment 1: The revised version of the manuscript did not include the response change indicated for line 154 under the Tissue handling and mRNA analysis section. It appears erroneously in line 169. Please correct.

AUTHOR RESPONSE: We appreciate the reviewer for identifying this discrepancy. The manuscript itself was revised as intended, but there was an error in our response to reviewer comment 1 in round 1. It should read:

"To ensure clarity, we have updated the manuscript in line 164: "Human tissue samples were analyzed using full mRNA sequencing, revealing differences in mRNA expression levels of NPC1L1, C7orf50, and GPR146."

We apologize for any confusion caused and have ensured that the correct information is reflected in the manuscript.